# Role of the C-Terminal β Sandwich of *Thermoanaerobacter tengcongensis* Thermophilic Esterase in Hydrolysis of Long-Chain Acyl Substrates

**DOI:** 10.3390/ijms25021272

**Published:** 2024-01-20

**Authors:** Enoch B. Joel, Adepeju Aberuagba, Adebayo J. Bello, Mariam Akanbi-Gada, Adedoyin Igunnu, Sylvia O. Malomo, Femi J. Olorunniji

**Affiliations:** 1School of Pharmacy & Biomolecular Sciences, Liverpool John Moores University, Byrom Street, Liverpool L3 3AF, UK; enjoebest@yahoo.com (E.B.J.); akanbigadamariam@gmail.com (M.A.-G.); 2Department of Biochemistry, Faculty of Basic Medical Sciences, University of Jos, Jos 930003, Nigeria; 3Department of Biological Sciences, McPherson University, Seriki-Sotayo 110117, Nigeria; 4Department of Biochemistry, Faculty of Life Sciences, University of Ilorin, Ilorin 234031, Nigeria; doyinigunnu@unilorin.edu.ng (A.I.);

**Keywords:** esterase, *Thermoanaerobacter tengcongensis*, C-terminal domain, substrate preference, thermophilic esterases

## Abstract

To search for a novel thermostable esterase for optimized industrial applications, esterase from a thermophilic eubacterium species, *Thermoanaerobacter tengcongensis* MB4, was purified and characterized in this work. Sequence analysis of *T. tengcongensis* esterase with other homologous esterases of the same family revealed an apparent tail at the C-terminal that is not conserved across the esterase family. Hence, it was hypothesized that the tail is unlikely to have an essential structural or catalytic role. However, there is no documented report of any role for this tail region. We probed the role of the C-terminal domain on the catalytic activity and substrate preference of *T. tengcongensis* esterase EstA3 with a view to see how it could be engineered for enhanced properties. To achieve this, we cloned, expressed, and purified the wild-type and the truncated versions of the enzyme. In addition, a naturally occurring member of the family (from *Brevibacillus brevis*) that lacks the C-terminal tail was also made. In vitro characterization of the purified enzymes showed that the C-terminal domain contributes significantly to the catalytic activity and distinct substrate preference of *T. tengcongensis* esterase EstA3. All three recombinant enzymes showed the highest preference for paranitrophenyl butyrate (pNPC4), which suggests they are true esterases, not lipases. Kinetic data revealed that truncation had a slight effect on the substrate-binding affinity. Thus, the drop in preference towards long-chain substrates might not be a result of substrate binding affinity alone. The findings from this work could form the basis for future protein engineering allowing the modification of esterase catalytic properties through domain swapping or by attaching a modular protein domain.

## 1. Introduction

Lipolytic enzymes (esterases and lipases) have received notable attention owing to their vast distribution in biological systems and environments and their significance in a wide range of industrial and biotechnological applications [1,2]. Their unique properties such as catalytic versatility, robustness, and high specificity have attracted enormous attention as industrial biocatalysts for a wide range of applications [3,4,5]. There is an increasing need for enzymes with extremophilic properties such as tolerance to high temperature, high salt concentrations, and stability in the presence of organic solvents—conditions associated with some downstream industrial processes [6,7]. Esterases and lipases displaying specificities for a broad range of triglycerides with varying acyl chain lengths are desirable for industrial processes designed for releasing free fatty acids for downstream applications. Examples of such applications inlude production of biodiesel and use in waste management processes that require breaking down lipid-based environmental pollutants [8]. Since these processes are usually carried at high temperatures, efforts at searching for enzymes that meet these criteria are often focused on thermophilic extremophile organisms.

An alternative approach is to engineer the properties of known esterases to acquire the features needed for the downstream processes where they are required. To achieve this, efforts are aimed at gaining a better understanding of the biochemical properties of esterases that could help in devising strategies for regulating and improving their catalytic efficiency and structural stability. 

Most lipases and esterases are relatively small, and they exhibit broad substrate specificities and not require any cofactors for their reactions. Therefore, the growing interest and demand, as well as the wide spectrum of applications have led to increased interest in the properties of these enzymes which could be exploited for other specific applications of esterase for optimized industrial and biotechnological processes [2].

Esterases and lipases belong to the α/β hydrolase superfamily, which exhibit common parallel β strands surrounded by α helical connections. Their active sites share characteristic sequence motif GXSXG pentapeptides and a catalytic triad consisting of serine-histidine-aspartic or glutamic acid [9].

*Thermoanaerobacter tengcongensis* esterase (LipA3, EstA3) is a thermophilic hydrolytic enzyme from a thermophilic eubacterium, *T. tengcongensis*, isolated from a hot spring in Tengchong, Yunnan Province, China, with high potential for industrial and biotechnological applications [10]. *T. tengcongensis* is an anaerobic, Gram-negative, rod-shaped bacterium that can survive in temperatures ranging from 50 to 80 °C [11]. *T. tengcongensis*, like other thermophilic microorganisms, usually survives in extreme environmental conditions like hot temperatures and can produce thermophilic enzymes with adaptation differences, structural differences, and higher hydrophobicity [12]. 

In an attempt to search for novel thermostable esterase for optimized industrial applications, a novel esterase gene, ORFs lipA (NP_622227) from a thermophilic eubacterium species, *T. tengcongensis* genome was reported by Zhang et al. [13]. *T. tengcongensis* esterase gene was heterologously expressed in *E. coli* and its biochemical properties characterized leading to its classification as the first member of the XIV family of lipolytic enzymes [10]. The biochemical characterization data also suggests that *T. tengcongensis* esterase is a promising candidate for industrial and biotechnological applications because of its inherent thermal stability. 

A remarkable variation in the primary sequence length among members of the XIV family of lipolytic enzymes (esterases) was reported (Figure 1) [10]. The sequence alignment of *T. tengcongensis* esterase with homologous esterases had earlier revealed an apparent tail of 106 non-conserved residues at the C-terminus with an unknown function [10]. The identified apparent tail at the C-terminal of *T. tengcongensis* esterase from sequence alignment of *T. tengcongensis* esterase with four other homologous esterases of the same family revealed that this region is not conserved across all family members. Multiple sequence analysis also revealed that the identified apparent tail at the C-terminal region of *T. tengcongensis* esterase is not conserved across all family members (Figure 1). Due to its lack of conservation across the family, it is unlikely that it plays an essential structural or catalytic role. An understanding of the role of the non-conserved C-terminal tail could provide some insight into how the properties of the enzyme could be engineered for improved activity. Specifically, we investigated if the C-terminal tail has a role in the catalytic efficiency of the enzyme or if the domain is a determinant of the enzyme’s specificity for substrates that differ in acyl chain length.

To achieve this objective, the wild-type *T. tengcongensis* esterase (EstA3_*Tt*) and the truncated version of the *T. tengcongensis* esterase, without the apparent tail (EstA3_*TtΔ*), as well as a naturally shorter version from *B. brevis* (EstA3_*Bb*) were synthesized, and their biochemical properties were compared. In vitro characterization of the activities of the three proteins reveals an interesting role for the C-terminal tail as a key determinant in binding and catalysis of esters with long-chain acyl groups. The implication of the findings for the mechanism of the enzyme and the potential application in protein engineering for applications in biotechnology are discussed.

## 2. Results

### 2.1. Alphafold-Predicted Structure of T. tengcongensis EstA3

Although *Thermoanaerobacter tengcongensis* esterase (EstA3_*Tt*) has been characterized and reported in the literature [10], there are no experimentally determined crystal structures in the database. Sequence alignment published by Rao et al. [10] suggests that the enzyme belongs to the α/β hydrolase superfamily of lipases (Figure 1). 

To better understand structure–function aspects of the enzyme, we searched through the AlphaFold Protein Database and downloaded the predicted structure of the enzyme (AF-Q8R7S7). Figure 2 shows a PyMOL rendering of the predicted structure. The sequence alignment of *T. tengcongensis* esterase with homologous esterases had earlier revealed an apparent tail containing 106 non-conserved residues at the C-terminus with an unknown function [10]. The C-terminal domain contains more than 50% hydrophobic residues that could be predicted to play some relevant role in the interaction with lipidic substrates as well as enzyme structural stabilization and regulation. Structure prediction using Phyre2 revealed a close structural similarity with *Bacillus licheniformis* lipase BlEst2, another member of the α/β hydrolase superfamily. An interesting feature is that the 106 amino-acid-residue C-terminal tail folds into a β sandwich that packs next to the catalytic domain. 

### 2.2. Design, Cloning, Expression, and Purification of Esterases

To probe the role of the C-terminal domain on the catalytic property and substrate preference of *Thermoanaerobacter tengcongensis* esterase, we expressed, purified, and compared the substrate preference and kinetic properties of the truncated enzyme against those of the full-length enzyme and a naturally shorter version from *B. brevis*. Probing the role of the non-conserved C-terminal domain (apparent tail) will allow a better understanding of its possible role in the catalytic activity, substrate selectivity and kinetics of these family proteins.

The three recombinant proteins containing 6xHis-tag were purified from the supernatants of cell lysates by single-step affinity chromatography on a Ni–NTA column [14]. Such high-level expression and efficient purification were similarly obtained for several esterases and other lipolytic enzymes using an *E. coli* expression system and Ni^2+^-affinity chromatography, as reported by Rao et al. [15], Jensen et al. [16], Latip et al. [17], and López-Fernández et al. [18]. Samples were taken during purification and assayed for esterase activity before and after dialysis, and all samples from the three purifications showed esterase activity. 

### 2.3. Hydrolytic Activities of Purified Esterases Reveal Different Substrate Preferences

Upon successfully achieving expression and purification and activity determination of the three recombinant enzymes (Table 1), it was necessary to assess the comparative substrate preference of the three recombinant enzymes. The concentrations of EstA3_Tt, EstA3_Tt∆, and EstA3_Bb recovered were 26.6, 27.4, and 28.09, respectively. All three preparations were concentrated as described in Materials and Methods and the final concentration of each prep was adjusted to 50 μM. 

The main objective of this study was to gain some insights into the role of the non-conserved C-terminal domain of a recombinant *T. tengcongensis* esterase, EstA3_*Tt* in distinct substrate preference. The activities of the three recombinant esterases on p-nitrophenyl acyl substrates (pNPC2, pNPC4, pNPC8, pNPC10, pNPC12, pNPC14, and pNPC16) are shown in Figure 3.

As shown in Figure 3, no significant variation was observed between the three enzymes in their hydrolysis of p-nitrophenyl esters with short chain lengths (pNPC2 and pNPC4), which suggests that the truncated C-terminal domain enhanced the recognition of long acyl chain substrates by the wild-type enzyme (EstA3_*Tt*), thereby broadening the substrate preference of the enzyme. Previous reports of López-Fernández et al. on a truncated prosequence of *Rhizopus oryzae* lipase also revealed the involvement of the 28 amino acids in the N-terminal domain of *Rhizopus oryzae* lipase in substrate specificity [18]. A noteworthy observation is that truncation of the C-terminus 106-amino-acid tail of EstA3_*Tt* resulted in a decrease in esterase catalytic hydrolysis of pNPC8 substrate by 4.2-fold, an indication that the tail may also be involved in the catalytic process through enhanced substrate binding during the hydrolysis reaction.

### 2.4. Kinetics of Activities of Purified Esterases on C4 and C8 Substrates

The kinetics of any enzyme are critical to understanding how enzymes behave in cellular systems and help predict the enzyme’s performance in industrial settings [19,20]. To further understand the role of the truncated C-terminal domain in the substrate utilization, the effects of varying substrate concentrations on the activities and kinetic parameters (K_m_, K_cat_, and V_max_) of the three enzymes-catalysed hydrolyses of short (pNPC4) and medium (pNPC8) acyl chain substrate were investigated and compared. Due to the lack of esterase activity of EstA3_*TtΔ* against substrates with an acyl chain longer than pNPC8, the kinetic study focused on only pNPC4 and pNPC8 substrates. 

All three recombinant esterases obeyed typical Michaelis–Menten kinetics for the tested substrates (Figure 4). The Lineweaver-Burk plot shows the dependence of the reciprocal of the initial rate on the reciprocal of the substrate concentration. The results of the effects of substrate concentration on the activities of the recombinant esterases show that the full-length enzyme (EstA3_*Tt*) was more active than the truncated and naturally shorter enzymes. Consistently, full-length enzyme EstA3_*Tt* performed better than the truncated and naturally shorter enzymes in all the tested substrates in the following order: EstA3_*Tt* > EstA3_*Bb* > EstA3_*TtΔ*. As summarized in Table 2, the results showed a decrease in maximum velocity and turnover rate with increasing chain length for all three recombinant esterases. 

In contrast, there were no significant changes in K_m_ values with increasing chain length for all three recombinant esterases (Table 2). The order of V_max_ for pNPC4 hydrolysis catalyzed by the enzymes was EstA3_*Tt* (263.1741 mM/min) > EstA3_*Bb* (256.41 mM/min) > EstA3_*TtΔ* (238.10 mM/min), while the order of substrate affinity (Km) was EstA3_*TtΔ* (0.64 mM) > EstA3_*Tt* (0.53 mM) = EstA3_*Bb* (0.53 mM). A similar trend of Vmax was observed for pNPC8 hydrolysis catalyzed by the three recombinant enzymes. On the other hand, the order of substrate affinity (Km) was EstA3*_Tt∆* (0.56 mM) > EstA3_*Bb* (0.51 mM) > EstA3_*Tt* (0.49 mM). The turnover number (K_cat_) of EstA3_*Tt* for pNPC4 substrate was approximately 2-fold and 1.9-fold higher than that of EstA3_*TtΔ* and EstA3_*Bb*, respectively. The K_cat_ value of EstA3_*Tt* (2.28 × 10^5^ s^−1^) for pNPC8 substrate was approximately 2.3- and 2.1-fold higher than that of EstA3_*TtΔ* (1.0 × 10^5^ s^−1^) and EstA3_*Bb* (1.1 × 10^5^ s^−1^), respectively. 

## 3. Discussion 

### 3.1. Substrate Preferences of EstA3_Tt, EstA3_Tt∆, and EstA3_Bb

*E. coli* is the most frequently used expression host system for laboratory and industrial scale production of enzymes and other proteins in biotechnology industry [21]. In this work, the custom-made recombinant plasmid DNA carrying esterase genes of interest were synthesized using the services of Invitrogen GeneArt (a gene synthesis commercial company). The three custom-built vector pET28a (+) with a T7 promoter from Invitrogen-GeneArt each carrying different esterase genes namely EstA3, EstA3∆, and EstA3.bb were transformed into *E. coli* BL21 (DE3) pLysS and a high-level expression was achieved in the first round of recombinant protein expression under predetermined optimization of cultivation and induction conditions. This indicates that all the three recombinant esterases (EstA3_*Tt*, EstA3_*TtΔ* and EstA3_*Bb*) were inherently amenable to the expression system used for this study. Ni-NTA affinity chromatography matrix allows rapid and efficient purification of recombinant proteins carrying a histidine tag [10,14,21].

The observation that all the three recombinant enzymes were more active on short acyl chain substrates (pNPC4) supports the previous findings that all the three recombinant enzymes are true esterases that favour the hydrolysis of short acyl-chain substrates (≤ C10) [9,22,23,24]. The three recombinant enzymes had the highest turn-over rate (Kcat) with the short-acyl chain substrate (pNPC4), and the highest affinities (low Km values) for long-acyl chain substrate. The observation that the esterase activity and substrate preference was affected by deletion of the truncated C-terminal domain with little effect on the substrate affinity (Km), suggests that such influence of the C-terminal domain may not be through facilitating substrate binding to the active site of the enzyme alone [25].

Our data from the catalytic activity, substrates preferences, and kinetics suggest that truncated C-terminal domain of non-conserved residues may contribute significantly to the catalytic activity and substrate promiscuity of EstA3_*Tt*. Martínez-Martínez et al. earlier predicted that the molecular mechanisms by which some esterases exhibit more substrate promiscuity than others remain unclear and yet to be elucidated [26]; findings from this study however, revealed that the non-conserved residues such as the ones depicted C-terminal domain could be one of the key contributory factors for the substrate promiscuity of the family of XIV of lipolytic enzymes (esterases) and possibly other esterases in general. This substrate promiscuity is an important and indisputable property of the esterase enzyme [9]. A narrow substrate spectrum is usually one of the most frequently encountered challenges for industrial applications of enzymes [27]. 

### 3.2. The C-Terminal Tail Contributes to the Stability of EstA3

Findings from catalytic properties suggest that the truncated C-terminal domain accounted for the drastic decrease in catalytic activity and instability. One possible explanation for this is that it is likely that the C-terminal domain plays a major role in folding of an enzyme to adopt its functionally active conformation during or following translation [21]. Our result implies that although the 106 amino acid residues at the C-terminal domain of EstA3 are not conserved across the family members, their absence in the truncated enzyme (EstA3_*TtΔ*) interferes with post-expression catalytic activity and stability. 

### 3.3. Role of the C-Terminal β Sandwich in Binding Long-Chain Acyl Substrates

Findings from this study suggest that the apparent tail (C-terminal domain) plays a critical role in substrate preference. The three enzymes displayed higher activity towards short-chain acyl substrates than the long-chain substrates, with the highest preference for p-nitrophenyl butyrate (pNPC4). The fact that the three recombinant lipolytic enzymes preferentially favor the hydrolysis of short-chain acyl ester (pNPC4) suggests they are classified as esterases rather than lipases [28]. A similar trend with pNPC4 as the most preferred substrate has also been reported for esterases from *Sulfolobus shibatae* [29], *Sulfolobus tokodaii* [30], *Bacillus subtilis* (RRL 1789) [31], and *Geobacillus* sp. HBB [32].

The preference of the three recombinant enzymes towards p-nitrophenyl substrates (pNPC2, pNPC4, pNPC8, pNPC10, pNPC12, pNPC14, and pNPC16) was established and showed that with increasing substrate acyl chain length, a corresponding decrease in esterase was observed. Furthermore, comparing the acyl-chain substrate utilization of the wild-type enzyme (EstA3_*Tt*) with the truncated enzyme (EstA3_*TtΔ*) and the naturally shorter enzyme (EstA3_*Bb*) revealed an abrupt and significant decrease in catalytic activity toward p-nitrophenyl esters with chain lengths ranging from C8 to C16. Such an abrupt decrease in the activity of the truncated enzyme toward p-nitrophenyl esters with chain lengths ranging from C8 to C16 was also reported for another naturally shorter esterase designated as EstEP16 [33]. EstEP16 was a new and thermostable esterase characterized from a metagenomic library derived from a deep-sea hydrothermal field in the east Pacific.

It is notable that the full-length enzyme (EstA3_*Tt*) hydrolyzed a broader spectrum of p-nitrophenyl esters (from C2 to C16) than the truncated enzyme (EstA3_*TtΔ*) and the naturally shorter enzyme (EstA3_*Bb*). Findings from the studies revealed that there was no detectable esterase activity towards acyl substrates bearing ≥ C10 for the truncated version of *T. tengcongensis* esterase. A similar result was reported for *Bacillus aryabhattai* esterase (BaCE) by Zhang et al. [30]. BaCE was active against esters with short-chain fatty acids (from pNPC2 to pNPC8) with high substrate specificity toward pNP butyrate (pNPC4), but no detectable activity was reported for long-chain acyl substrates (≥C10) [34]. 

The β sandwich sequence did not retrieve any structure from the NCBI Conserved Domain Architecture Retrieval tool (https://www.ncbi.nlm.nih.gov/Structure/lexington/lexington.cgi, accessed on 1 January 2024). We ran the predicted sandwich through the Dali server (http://ekhidna2.biocenter.helsinki.fi/dali/, accessed on 1 January 2024), and most of the structural matches were to integrins. However, the closest match was a crystal structure (6WPX) of *Bacillus licheniformis* lipase BlEst2 in propetide form [35]. Structure prediction using Phyre2 identified a close structural alignment of LipA3 with the C-terminal domain of BlEst2, an indication that this feature is common among members of the α/β hydrolase superfamily. It is likely that the β sandwich is involved in binding the hydrophobic part of the long-chain fatty acid moiety since its deletion resulted in loss of activity of C12–16 substrates. Due to the high hydrophobicity of the tail region, it is conceivable that the domain could facilitate interaction of lipidic substrates, hence its contribution to the ability of EstA3_*Tt* to bind and hydrolyze long-chain fatty acids. 

Similar observations have been made in other lipases in which hydrophobic domains are attributed to catalytic specificity for long-chain substrates [18,36]. The findings from this work, coupled with recent developments in protein structure prediction and modelling, can be used to alter the specificity and catalytic efficiency of the enzyme, as was done recently for a *Rhizopus oryzae* lipase [37].

Findings from the substrate preference studies indicate that the non-conserved C-terminal domain of EstA3_*Tt* plays a noticeable role in selective substrate preference, especially towards long-chain acyl substrates. Hence, the widely accepted assertion that the variable sequence/non-conserved sequence does not play a catalytic role in a group or family of enzymes, may not be the case for lipolytic enzymes. The involvement of the truncated non-conserved C-terminal domain in the catalytic activity and substrate preference of EstA3_*Tt*, as revealed in this work, matches the findings from Kovacic et al. that non-catalytic residues may contribute notably to substrate specificity, protein activity, and stability through yet-to-be elucidated mechanisms [38]. Although this finding is relevant to the XIV family of lipolytic enzymes (esterases), it has provided new openings for deeper research to understand the actual role and mechanism of non-conserved residues of the general lipolytic enzymes (esterases and lipases). More subtle experimental designs and targeted mutagenesis studies are required to gain a better understanding of how these systems work and may reflect a new paradigm in the regulation of enzyme activity. 

The proposed contribution of the C-terminal domain to broad substrate preference could open a range of industrial applications for esterases. Therefore, such observed broad substrate preference implies that the full-length enzyme could also perform some specific industrial applications where lipases are usually used in the catalysis of medium- and long-chain acyl substrates. This might consequently reduce the cost of producing multiple enzymes for different industrial applications. Acquisition of new specificities without compromising existing ones is the major driving force for the natural evolution of enzymes with novel specificities through microbial adaptation to extreme ecosystems [39]. The presence of the C-terminal domain in the wild-type enzyme (EstA3_*Tt*) conferred a broad substrate spectrum, whereas its absence narrowed the substrate spectrum of the truncated enzyme (EstA3_*TtΔ*). The molecular mechanism by which the C-terminal domain facilitates more activity towards the long-chain acyl substrates remains unknown and needs further investigation.

### 3.4. Structure-Guided Engineering of Esterases Could Generate New Catalytic Flexibility for Industrial Applications 

Exploiting protein engineering in attaching the C-terminal domain to other naturally short esterase family members broadens their industrial applications, where long-chain acyl lipids are used as substrates. On the other hand, since the truncation did not massively affect esterase activity towards short acyl esters pNPC2 and pNPC4, industrial applications that require hydrolysis of short-chain acyl substrates might prefer truncated esterase enzymes to cut the cost of production. 

The overall objective of this study was to decipher the outcome of truncation of an apparent tail sequence with non-conserved residues, and findings from the experimental data revealed the involvement of the truncated C-terminal domain in the catalytic activity and preference towards long-chain substrates. The truncation resulted in a decrease in esterase catalytic activity by 4.2-fold. The full-length enzyme showed more preference for long-chain acyl substrates compared to two short-length enzymes, which indicate a significant role of the truncated C-terminal domain in broad substrate preference.

The findings represent a significant improvement in our understanding and suggest that the widely accepted trademark about the involvement of the non-conserved residues/sequence in enzyme catalysis may not necessarily be the case in esterases. This study particularly provides a material basis for the role of non-conserved sequences in catalytic activity and substrate preference among members of the XIV family of lipolytic enzymes, which can be explored for more efficient industrial applications. This study also provides useful information that would spur curiosity for further research to better understand the mechanism through which the truncated apparent C-terminal domain contributes to the observed properties of *T. tengcongensis* esterase. 

The findings from this work could form the basis for future protein engineering allowing the modification of esterase catalytic properties through domain swapping or by attaching a modular protein domain. More subtle experimental designs and targeted mutagenesis studies would be needed to gain a better understanding of how these systems work and may reflect a new paradigm in the regulation of enzyme activity. Studies on the effects of pH and temperature on reaction rates could provide more insights into the factors that regulate the activities of the esterases and the role of the C-terminal tail. The understanding of the role played by the non-conserved C-terminal domain in determining the properties of this enzyme could open new insight into the effective utilization of esterases and other lipolytic enzymes for optimized industrial and biotechnological applications. 

Further studies are required to probe the significance of the acquisition of the C-terminal domain in the adaptive biology and/or localization of the protein.

## 4. Materials and Methods

### 4.1. Plasmids and Bacterial Strains 

*E. coli* BL21(DE3) plysS host strain (Novagen, Germany), pET-28a (+) (Novagen), the protein overexpression vector, and all other reagents for molecular cloning were made available through the research group of Dr. Femi Olorunniji, School of Pharmacy and Biomolecular Science, Liverpool John Moores University, UK. Gene sequences were obtained from the National Centre for Biotechnology Information (NCBI) database and codon-optimized. Gene synthesis and custom-made plasmids, each carrying different esterase genes, namely FEM45 (EstA3_*Tt*), FEM103 (EstA3_*Tt∆*), and FEM170 (EstA3_*Bb*), were constructed by Synthetic Biology Company Invitrogen GeneArt, Thermo Fisher Scientific, Cambridge, UK.

### 4.2. Chemicals

Para-nitrophenyl esters (C2-C16), p-nitrophenyl acetate (pNPC2), p-nitrophenyl butyrate (pNPC4), p-nitrophenyl caprylate (pNPC8), p-nitrophenyl caprate (pNPC10), p-nitrophenyl laurate (pNPC12), p-nitrophenyl myristate (pNPC14), p-nitrophenyl palmitate (pNPC16), isopropyl-β-D-thiogalactopyranoside (IPTG), chloramphenicol, kanamycin, and all other reagents were obtained from Sigma (Hampshire) and New England Biolab, Hertfordshire, (UK) and were of analytical grade.

### 4.3. Protein Expression and Purification

The custom-made plasmids, each carrying different esterase genes, namely pFEM45 (EstA3_*Tt*), pFEM103 (EstA3_*TtΔ*), and pFEM170 (EstA3_*Bb*), were transformed into *E. coli* BL21 (DE3) pLysS cells. 

The transformed *E. coli* BL21(DE3) pLysS cells containing custom-made recombinant plasmids were cultured in Luria–Bertani (LB) medium (pH 7.0) containing 100 µL of µg/mL Chloramphenicol and 100 µL of 50 µg/mL Kanamycin with constant shaking at 225 rpm, 37 °C to mid log phase (when the cell density at OD_600_ reached 0.8). Isopropyl thio-β-D-galactoside (IPTG) was added (at the final concentration of 1 mM) to the cell cultures, which were allowed to grow overnight with constant shaking at 225 rpm, 20 °C in a refrigerated incubator (EES-60 Model). Induced cells were harvested by centrifugation at 5000× *g* for 10 min using a refrigerated centrifuge at 4 °C. The cell pellets were resuspended and incubated in 50 mM Tris–HCl buffer (pH 8.0) at 4 °C. The cell suspension was then lysed by ultrasonication using a Branson Sonifier^®^, Model-250 for 5 min (10 sec ON/OFF pulses). After ultrasonic cell disintegration, the cell debris was removed by centrifugation (4000 rpm, 30 min). Subsequently, the supernatant was purified with a Nickel affinity column (Ni-his NTA Novagen). Supernatant was loaded on Nickel affinity column and eluted using a high concentration of imidazole. The fractions containing esterase activity were collected and concentrated by dialysis against glycerol storage/dialysis Tris-HCl pH buffer 7.5 (made up of a final concentration of the following composition: 25 mM Tri-HCl pH 7.5; 50% Glycerol; 1 mM DTT; 1000 mM NaCl) overnight with constant shaking using a magnetic stirrer in a cold room. The Nickel affinity chromatography purification steps were carried out using an ÄKTA Purifier according to standard protocol. The recovered proteins were further concentrated using Pierce Protein Concentrator columns (Thermofisher Scientific) and the stock concentrations were adjusted to 50 μM.

### 4.4. Determination of Esterase Activity

Protein concentration was determined by measuring absorbance at 280 nm. The concentrations of the three proteins in μg/mL and μM are as shown in Table 1. The molecular weight, A_280_ molar extinction coefficient, and A_280_ extinction coefficient (1000 μg/mL) for all three proteins were calculated using the EMBL-EBI online tool EMBOSS Pepstats (https://www.ebi.ac.uk/Tools/seqstats/emboss_pepstats/, accessed on 1 January 2024).

Enzyme activity was monitored spectrophotometrically at 405 nm by using a CLARIOstar plate reader (BMG LABTECH) with p-nitrophenyl esters (pNPC2, pNPC4, pNPC8, pNPC10, pNPC12, pNPC14, and pNPC16). The activities of the recombinant esterases were determined in reactions that contained 50 mM phosphate buffer pH 7.4, 20 mM p-nitrophenyl ester and esterase enzyme. An aliquot of 10 μL of 20 mM of the appropriate p-nitrophenyl ester was added to 100 μL of 50 mM Tris-HCl buffer (pH 8.5) and 70 μL of distilled water to form a 180 μL “buffered substrate mixture”. Reactions were initiated by the addition of 20 μL of enzyme (50 μM) to “buffered substrate mixture” and incubated at 50 °C for 15 min. The change in A_405_ was monitored immediately and continuously after starting the reaction over 15 min in a CLARIOstar Plate Reader (BMG LABTECH) using an in-built thermostat. Absorbance changes were converted to the amount of p-nitrophenol formed using a molar extinction coefficient of 18000 M^−1^cm^−1^. The initial reaction rate, V_o_, expressed as mM/min, and the amount of p-nitrophenol formed after 15-min reaction, expressed as millimoles (mM), were determined from the time course of the reaction. The initial rates (Vo) were calculated from the slope of the initial (5 min) linear portion of the increase in absorbance.

### 4.5. Determination of Substrate Preference and Kinetic Parameters

Substrate specificity of the enzyme was determined using various p-nitrophenyl esters with different acyl chains (pNPC2, pNPC4, pNPC8, pNPC10, pNPC12, pNPC14 and pNPC16). The kinetic parameters (K_m_ and V_max_) of the recombinant esterases using short- and medium-chain substrates (pNPC4 and pNPC8) were determined. The plot of esterase activity (V) versus substrate concentration ([S]) (Michaelis–Menten plot) was used to show the kinetic behaviors of the esterases. On the other hand, the equation of the straight line of the reciprocal plot of esterase activity (V) against the substrate concentration ([S]) (Lineweaver–Burk plot) was used to calculate the kinetic parameters (K_m_ and V_max_). The turnover number (K_cat_) was calculated as Vmax divided by the corresponding enzymatic enzyme concentration.

## Figures and Tables

**Figure 1 ijms-25-01272-f001:**
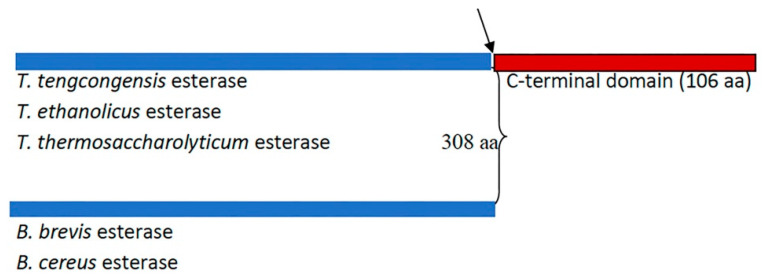
Schematic representation of the number of amino acid residues of *Thermoanaerobacter tengcongensis* esterase (EstA3_*Tt*) sequence and four homologous proteins. The blue shaded colour indicates the region of the sequence of conserved residues. The maroon shade colour indicates the apparent tail of non-conserved residues found in three homologous proteins (*T. tengcongesis, T. ethanolicus,* and *T. thermosaccharolyticum*) and absent in the other two homologous proteins (*B. brevis, B. cereus*). The black arrow indicates the point of truncation of *Thermoanaerobacter tengcongensis* esterase to make a shorter version of the same length of amino acid sequence (EstA3_*TtΔ*) with the naturally shorter version from *B. brevis,* (EstA3_*Bb*).

**Figure 2 ijms-25-01272-f002:**
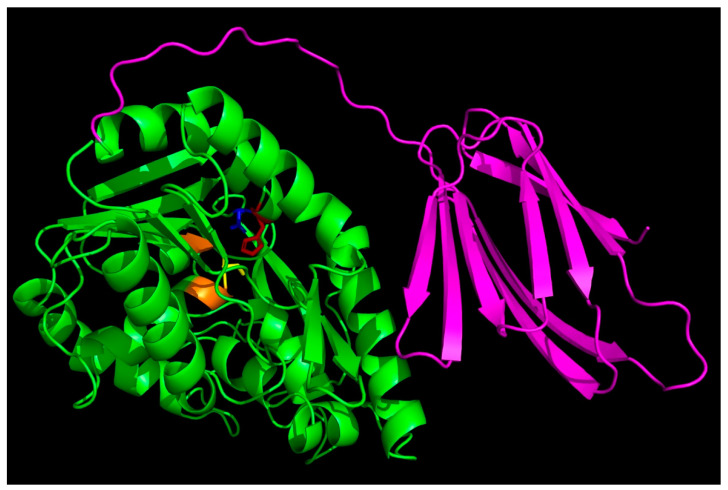
AlphaFold-predicted structures of *Thermoanaerobacter tengcongensis* esterase. The PDB files (AF-Q8R7S7-F1) were downloaded from the AlphaFold Protein Structure Database (https://alphafold.ebi.ac.uk/, accessed on 1 January 2024), accessed on 30 December 2023, and visualized with PyMOL (https://pymol.org/2/, accessed on 1 January 2024). The C-terminal tail is shown in magenta, and the side chains of the key active site residues S92, D270, and H292 are shown in yellow, blue, and red sticks, respectively.

**Figure 3 ijms-25-01272-f003:**
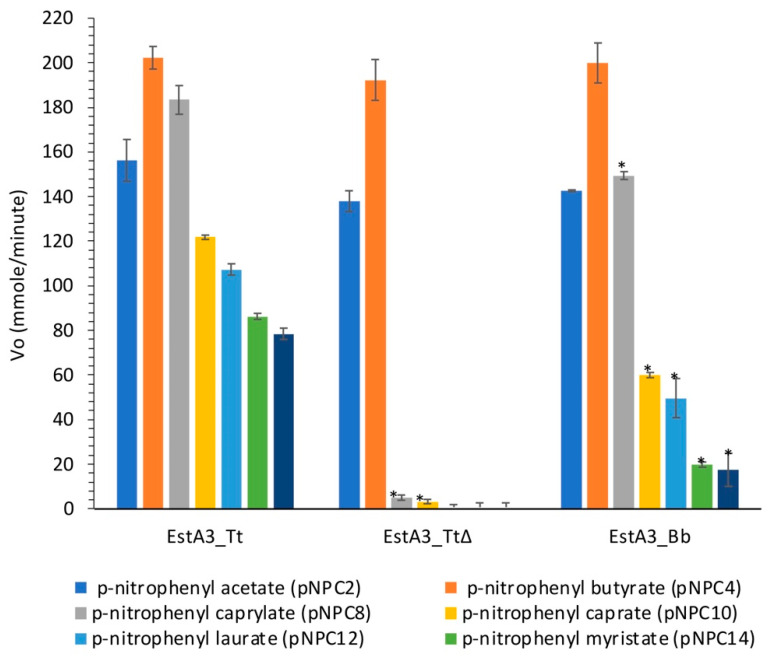
Substrate preference of the wild-type *Thermoanaerobacter tengcongensis* esterase (EstA3_*Tt*), truncated *Thermoanaerobacter tengcongensis* esterase (EstA3_*TtΔ*), and *Brevibacillus brevis* esterase (EstA3_*Bb*). Buffered substrate mixture composed of 20 mM of the substrate (pNPC2, pNPC4, pNPC10, pNPC12, pNPC14, and pNPC16), distilled water, and 50 mM sodium phosphate buffer pH 7.4 at a ratio of 1:7:10 (*v*/*v*/*v*), respectively. Each reaction was started by incubating 20 μL (50 μM) of the three recombinant esterases (5 μM final concentration) with 180 μL of a buffered substrate mixture of different substrates (pNPC2, pNPC4, pNPC8, pNPC10, pNPC12, pNPC14, and pNPC16) for 15 min at 50 °C. Values are expressed as mean ± standard deviation of results from three independent experiments. Bars with asteriks for each enzyme are significantly different (*p* < 0.05) from those without the asterisk. EstA3_*Tt*: wild-type *T. tengcongensis* esterase; EstA3_*TtΔ*: truncated mutant of *T. tengcongensis* esterase; EstA3_*Bb*: a natural shorter version of an esterase from *B. brevis*.

**Figure 4 ijms-25-01272-f004:**
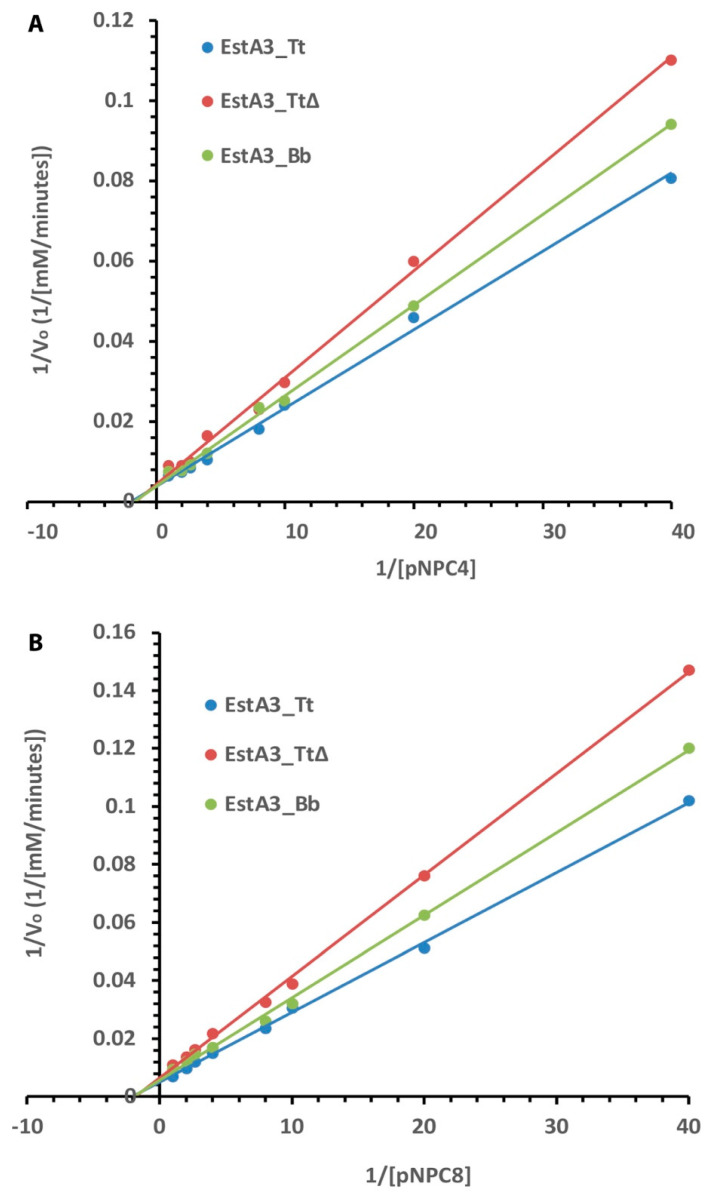
Lineweaver–Burk plot of the effect of substrate concentrations on the activities of EstA3_*Tt*, EstA3_*Tt*∆, and EstA3_*Bb*. (**A**) p–nitrophenyl butyrate (pNPC4); (**B**) p–nitrophenyl caprylate (pNPC8). Buffered substrate mixture composed of 20 mM of the substrate, distilled water, and 50 mM sodium phosphate buffer pH 7.4 at a ratio of 1:7:10 (*v*/*v*/*v*), respectively. Activities of the three recombinant esterases at 5 μM final concentration were measured at varying concentrations (1–20 mM) of pNPC4. (**A**) or pNCP8 (**B**). Each reaction was initiated by incubating 20 μL (50 μM) of enzymes with 180 μL of buffered substrate and incubated at 50 °C for 15 min. Activities were determined using different concentrations (1.0, 1.25, 2.5, 5.0, 10, 15, and 20 mM) of pNPC4 (**A**) or pNCP8 (**B**). The reciprocals of the reaction initial velocities and the substrate concentrations were used for the plot, and values are expressed as means of results from three independent experiments.

**Table 1 ijms-25-01272-t001:** Protein concentrations of purified EstA3_Tt, EstA3_Tt∆, and EstA3_Bb.

Enzymes	EstA3_Tt	EstA3_Tt∆	EstA3_Bb
Molecular weight	47,078	34,719	35,243
Molar Extinction Coefficient (M^−1^cm^−1^)	71,280	58,330	48,820
A_280_ (1000 μg/mL)	1.514	1.680	1.385
Protein Concentration (μg/mL)	1250	950	990
Protein Concentration (μM)	26.6	27.4	28.09

**Table 2 ijms-25-01272-t002:** Kinetic parameters of the wild-type *Thermoanaerobacter tengcongensis* esterase, truncated *Thermoanaerobacter tengcongensis* esterase, and *Brevibacillus brevis* esterase for the hydrolyses of various p-nitrophenyl esters. Kinetic parameters (K_m_ and V_max_) were calculated from the equation of the straight line of the Lineweaver–Burk plots shown in Figure 4.

Enzymes	Km (mM)	Vmax (mM/Minutes)	K_cat_ × 10^5^ (S^−1^)
	pNPC4		
EstA3_Tt	0.53	263.17	2.94
EstA3_Bb	0.59	256.41	1.51
EstA3_Tt∆	0.64	238.10	1.50
	pNPC8		
	0.49	204.08	2.28
EstA3_Tt	0.51	181.82	1.07
EstA3_Bb	0.56	158.73	1.00

Kinetic parameters (K_m_ and V_max_) were calculated from the equation of the straight line of the Lineweaver–Burk plots shown in Figure 4.

## Data Availability

Data is contained within the article.

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
