# Peer review of "Role of the C-Terminal β Sandwich of Thermoanaerobacter tengcongensis Thermophilic Esterase in Hydrolysis of Long-Chain Acyl Substrates"

_ijms, 2024, doi:10.3390/ijms25021272_

Round 1

Reviewer 1 Report

Comments and Suggestions for Authors

Well written and no obvious technical deficiencies.  This is a solid piece of work, and may prove to be impactful, although in current form it leaves the functional implications of the discovery a bit vague.

The following are a number of prompts that are all optional, but might enhance the insight presented by the paper:

Is there evidence of tail cleavage that might alter localization and/or function?

Does the T. tengcongensis esterase exhibit significant localization or function differences relative to that of B. brevis?

Esterase variation frequently corresponds to environmental pressures, including frequent examples of pesticide-induced adaptation.  Are there any potential stimuli that might motivate the development in T. tengcongensis of a novel variant?

I applaud the effort to characterize the C-terminal structure/function through exploration of the Conserved Domain Architecture site, and Dali, but there are other prospective analyses that might augment this insight.  Taken completely out of the esterase context, I would be very surprised if the predicted C-terminal beta-sandwich did not have other functional analogs identifiable through the many threading tools out there.  These might be worth a look.

Author Response

Reviewer 1

Comment 1: Well written and no obvious technical deficiencies.  This is a solid piece of work, and may prove to be impactful, although in current form it leaves the functional implications of the discovery a bit vague.

Response to Comment 1: We that the Reviewer for their very positive assessment of the manuscript and the suggestion that it may be impactful in the field. The functional implication of the discovery is primarily fundamental in nature giving us an insight into how an additional domain could confer novel or extended functionality to a protein. In addition, the finding could form the basis for future protein engineering allowing the modification of esterase catalytic properties through domain swapping or attaching a modular protein domain that brings a different property. These points are discussed in Sections 3.3 and 3.4. We have made some changes to the Abstract, the Introduction, and the relevant parts of the Discussion to further highlight these points.

As mentioned in the response to the specific points below, we agree with the Reviewer that a more holistic discussion of the significance of the C-terminal domain in the adaptive biology and/or localization of the protein would be interesting. However, the data presented in this work cannot be interpreted to answer those questions. We appreciate the Reviewer’s point on this, and we will investigate these points in future work.

The following are a number of prompts that are all optional, but might enhance the insight presented by the paper:

Comment 2: Is there evidence of tail cleavage that might alter localization and/or function?

Response to Comment 2: We appreciate the suggestion from the Reviewer to investigate this possibility. We have not done any experiment that might provide a hint at the role of the tail during translation and localisation of the protein. There is also no direct evidence from the literature to that effect.

Comment 3: Does the T. tengcongensis esterase exhibit significant localization or function differences relative to that of B. brevis?

Response to Comment 3: Again, we appreciate the suggestion from the Reviewer to explore differences in the biological functions of the two esterases. Our approach in this report is to highlight the likely function of the C-terminal tail on the in vitro activities of the protein. We have not investigated any inherent biological role of the proteins in their respective organisms.

Comment 4: Esterase variation frequently corresponds to environmental pressures, including frequent examples of pesticide-induced adaptation.  Are there any potential stimuli that might motivate the development in T. tengcongensis of a novel variant?

Response to Comment 4: T. tengcongensis is a thermophilic organism. Hence, it is not surprising that the esterase manifests thermophilic properties. It is not unusual for extremophilic organisms to adapt in response to multiple environmental pressures. However, we have not taken an approach that will answer this question. We agree with the Reviewer that a broader contextual discussion of the role of the C-terminal domain in the overall biology of the organism would be interesting. We will make that a subject for further investigation and will hopefully report the findings in another paper.

Comment 5: I applaud the effort to characterize the C-terminal structure/function through exploration of the Conserved Domain Architecture site, and Dali, but there are other prospective analyses that might augment this insight.  Taken completely out of the esterase context, I would be very surprised if the predicted C-terminal beta-sandwich did not have other functional analogs identifiable through the many threading tools out there.  These might be worth a look.

Response to Comment 5: We thank the Reviewer for the suggestion to look into other approaches to improve our insight into the structure/function of the C-terminal domain. In addition to the ones reported in the manuscript, we also used the Foldseek server (https://search.foldseek.com/search) and we got results like what the other systems returned. We thought it was not necessary to list all the searches carried out. We have also taken a conservative approach in interpreting the predicted structure in the context of the experimental data that we have.

Reviewer 2 Report

Comments and Suggestions for Authors

The MS submitted by Enoch B. Joel and coworkers aims the characterization of the role of the C-terminal β sandwich of Thermoanaerobacter  tengcongensis thermophilic esterase in hydrolysis of long chain 3 acyl substrates. This enzyme has potential uses in biotechnology and the knowledge of the catalytic efficiency of this enzyme could support its use in the future in some industrial processes.

It is a work mainly focused on the biochemical characterization of the modified enzyme in which the C-ter is truncated. This modified enzyme is compared to the wild type and the one from a Bacillus species.

The work is in general well written and organised. However, some minor issues must be addressed by the authors. Comments have been embedded through the MS to help the authors to improve this version.

Hope you find it useful.

Comments on the Quality of English Language

Minor typos and grammar issues should be revised.

Author Response

Reviewer 2

Comment: The MS submitted by Enoch B. Joel and coworkers aims the characterization of the role of the C-terminal β sandwich of Thermoanaerobacter  tengcongensis thermophilic esterase in hydrolysis of long chain 3 acyl substrates. This enzyme has potential uses in biotechnology and the knowledge of the catalytic efficiency of this enzyme could support its use in the future in some industrial processes.

It is a work mainly focused on the biochemical characterization of the modified enzyme in which the C-ter is truncated. This modified enzyme is compared to the wild type and the one from a Bacillus species.

The work is in general well written and organised. However, some minor issues must be addressed by the authors. Comments have been embedded through the MS to help the authors to improve this version.

Hope you find it useful.

Response to Comment: We appreciate the positive comments of the Reviewer on the structure and organisation of the manuscript, and the confidence expressed that the enzyme could be useful in some industrial processes. We also appreciate the changes made by the Reviewer in the version sent by the editorial office for this revision.